# SAM.MD: Zero-shot medical image segmentation capabilities of the *Segment Anything Model*

**Saikat Roy**[*1]                                                    SAIKAT.ROY@DKFZ-HEIDELBERG.DE

**Tassilo Wald**[*1,2]                                               TASSILO.WALD@DKFZ-HEIDELBERG.DE

**Gregor Koehler**[*1,2]                                              G.KOEHLER@DKFZ-HEIDELBERG.DE

**Maximilian R. Rokuss**[*1]                            MAXIMILIAN.ROKUSS@DKFZ-HEIDELBERG.DE

**Nico Disch**[*1]                                                    NICO.DISCH@DKFZ-HEIDELBERG.DE

**Julius Holzschuh**[*1]                                 JULIUS.HOLZSCHUH@DKFZ-HEIDELBERG.DE

**David Zimmerer**[*1]                                            D.ZIMMERER@DKFZ-HEIDELBERG.DE

**Klaus H. Maier-Hein**[1,3]                                  K.MAIER-HEIN@DKFZ-HEIDELBERG.DE

[1] *Medical Image Computing, German Cancer Research Center (DKFZ), Heidelberg, Germany*

[2] *Helmholtz Imaging*

[3] *Pattern Analysis and Learning Group, Heidelberg University Hospital, Germany*

**Editors:** Under Review for MIDL 2023

## Abstract

Foundation models have taken over natural language processing and image generation domains due to the flexibility of prompting. With the recent introduction of the Segment Anything Model (SAM), this prompt-driven paradigm has entered image segmentation with a hitherto unexplored abundance of capabilities. The purpose of this paper is to conduct an initial evaluation of the out-of-the-box zero-shot capabilities of SAM for medical image segmentation, by evaluating its performance on an abdominal CT organ segmentation task, via point or bounding box based prompting. We show that SAM generalizes well to CT data, making it a potential catalyst for the advancement of semi-automatic segmentation tools for clinicians. We believe that this foundation model, while not reaching state-of-the-art segmentation performance in our investigations, can serve as a highly potent starting point for further adaptations of such models to the intricacies of the medical domain.

**Keywords:** medical image segmentation, SAM, foundation models, zero-shot learning

## 1. Introduction

In recent years, there has been an explosion in the development and use of foundational models in the field of artificial intelligence. These models are trained on very large datasets in order to generalize on various tasks and domains. In the Natural Language Processing domain, Large Language Models (LLMs) have taken over (Brown et al., 2020). This leads to models of increasing size culminating in the recent GPT4 by OpenAI (2023). For the image domain, Stable Diffusion (Rombach et al., 2022) and DALL-E (Ramesh et al., 2021), are models that generate high-resolution images using text prompts. And with the recent publication of the Segment Anything Model (SAM) (Kirillov et al., 2023) the field of image segmentation received a promptable model, possibly enabling a wide range of applications. In this paper, we contribute an early stage evaluation of SAM with different visual prompts demonstrating varying degrees of accuracy on a multi-organ dataset of the CT domain.

---

[*] Contributed equally

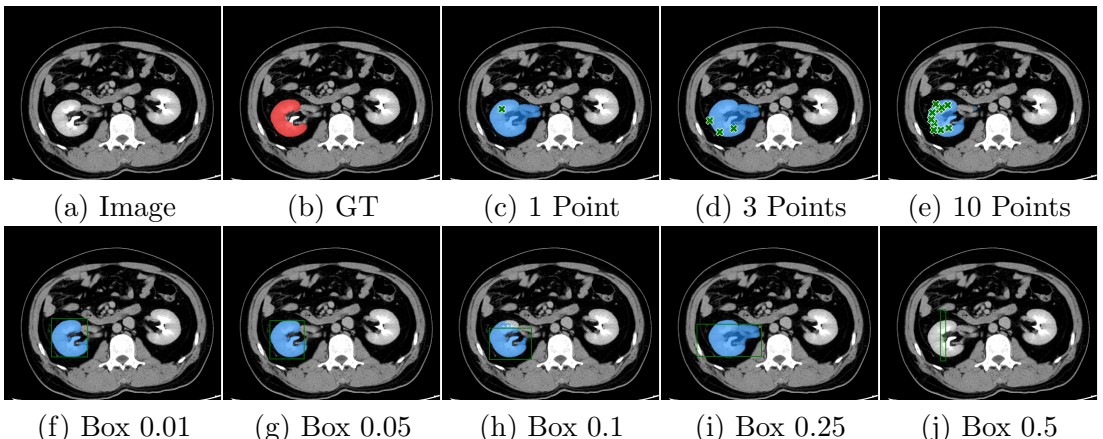

Figure 1: Examples of random point and jittered box prompts with subsequently generated segmentation masks. Prompt points and boxes are represented in green, while the obtained segmentations are shown in blue.

## 2. Methods

**Slice Extraction**   We use slices from the AMOS22 Abdominal CT Organ Segmentation dataset (Ji et al., 2022) to evaluate the zero-shot capabilities of SAM. We generate our evaluation dataset using axial 2D slices of patients centered around the center-of-mass of each given label. This results in 197-240 slices *per patient, per class* with each image slice containing some foreground class and a corresponding binary mask. Given this slice and binary mask, we generate different types of visual prompts.

**Visual Prompt Engineering**   Zero-shot approaches have recently utilized prompting to segment novel concepts or classes not seen during training (Lüddecke and Ecker, 2022; Zou et al., 2022). SAM allows a variety of prompts including text, points and boxes to enable zero-shot semantic segmentation.[1] In this work, we use the following *limited set* of *positive* visual prompts to gauge the zero-shot capabilities of SAM on unseen concepts – 1) Point-based prompting with 1, 3 and 10 randomly selected points from the segmentation mask of the novel structure, 2) Bounding boxes of the segmentation masks with jitter of 0.01, 0.05, 0.1, 0.25 and 0.5 added randomly, to simulate various degrees of user inaccuracy. Boxes and Points are provided in an Oracle fashion to imitate an expert clinician.

## 3. Results and Discussion

### 3.1. Results

We compare the predictions of SAM to the corresponding 2D slices extracted from predictions of a trained 2D and 3D nnU-Net baseline (Isensee et al., 2018). Dice Similarity Coefficient (DSC) of the various prompting types as well as nnU-Net are shown in Table

---

1. To the best of our knowledge, SAM does not provide a direct text prompt interface yet.

| Method | Organs | | | | | | | | | | | | | | AVG | AVG* |
|---|---|---|---|---|---|---|---|---|---|---|---|---|---|---|---|---|
| | Spl. | R.Kid. | L.Kid. | GallBl. | Esoph. | Liver | Stom. | Aorta | Postc. | Pancr. | R.AG. | L.AG. | Duod. | Blad. | | |
| 1 Point | 0.632 | 0.759 | 0.770 | 0.616 | 0.382 | 0.577 | 0.508 | 0.720 | 0.453 | 0.317 | 0.085 | 0.196 | 0.339 | 0.542 | 0.493 | 0.347 |
| 3 Points | 0.733 | 0.784 | 0.786 | 0.683 | 0.448 | 0.658 | 0.577 | 0.758 | 0.493 | 0.343 | 0.129 | 0.240 | 0.325 | 0.631 | 0.542 | 0.397 |
| 10 Points | 0.857 | 0.855 | 0.857 | 0.800 | 0.643 | 0.811 | 0.759 | 0.842 | 0.637 | 0.538 | 0.405 | 0.516 | 0.480 | 0.789 | 0.699 | 0.560 |
| Boxes, 0.01 | 0.926 | 0.884 | 0.889 | 0.883 | 0.820 | 0.902 | 0.823 | 0.924 | 0.867 | 0.727 | 0.618 | 0.754 | 0.811 | 0.909 | 0.838 | 0.826 |
| Boxes, 0.05 | 0.920 | 0.883 | 0.894 | 0.879 | 0.814 | 0.883 | 0.818 | 0.923 | 0.862 | 0.727 | 0.609 | 0.746 | 0.805 | 0.907 | 0.834 | 0.819 |
| Boxes, 0.1 | 0.890 | 0.870 | 0.874 | 0.859 | 0.806 | 0.813 | 0.796 | 0.919 | 0.845 | 0.702 | 0.594 | 0.733 | 0.785 | 0.862 | 0.810 | 0.795 |
| Boxes, 0.25 | 0.553 | 0.601 | 0.618 | 0.667 | 0.656 | 0.490 | 0.561 | 0.747 | 0.687 | 0.481 | 0.478 | 0.558 | 0.655 | 0.561 | 0.594 | 0.612 |
| Boxes, 0.5 | 0.202 | 0.275 | 0.257 | 0.347 | 0.356 | 0.164 | 0.252 | 0.381 | 0.335 | 0.239 | 0.234 | 0.308 | 0.343 | 0.205 | 0.278 | 0.289 |
| nnUNet 3D | 0.978 | 0.951 | 0.951 | 0.903 | 0.856 | 0.978 | 0.919 | 0.961 | 0.923 | 0.856 | 0.790 | 0.815 | 0.814 | 0.929 | 0.902 | 0.902 |
| nnUNet 2D | 0.977 | 0.938 | 0.943 | 0.865 | 0.850 | 0.976 | 0.890 | 0.954 | 0.884 | 0.788 | 0.753 | 0.787 | 0.745 | 0.920 | 0.877 | 0.877 |

Table 1: DSC of Point and Box Prompting against 2D and 3D nnUNet. All results created after CT clipping to -100 to 200 Hounsfield Units, except **AVG\*** on the right which is the average DSC on raw CT values.

1. Box prompting, even with moderate (0.1) jitter, is seen to be highly competitive against our baselines, compared to Point prompts.

## 3.2. Discussion

**Zero-shot Medical Image Segmentation**   SAM is seen to segment novel target structures (organs), especially with bounding box prompting at moderate jitter, to highly competitive accuracies compared to our baselines. Single positive bounding boxes are seen to perform considerably better than 10 positive point prompts. The performance does not degrade on raw CT values as well (AVG\*), indicating robustness of box prompting to high intensity ranges. Considering that nnU-Net is a strong automatic baseline trained on the entire dataset and SAM only sees a slice and a prompt (points or box), SAM demonstrates enormous potential as a zero-shot technique for medical image segmentation.

**Who is it useful for?**   Our experiments demonstrate that SAM could be highly beneficial for interactive segmentation pipelines – enabling rapid semi-automatic segmentation of a majority of the structure of interest, with only a few click or bounding box prompts (or possibly both) by an expert. Empirically, it appears that SAM may experience decreased accuracy in areas near class boundaries (as shown in Figure 1). However, as such areas can be manually segmented, the use of SAM might still greatly improve the speed of a clinical pipeline while maintaining a good level of accuracy.

## 4. Conclusion

Our study evaluates the zero-shot effectiveness of the Segment Anything Model (SAM) for medical image segmentation using few click and bounding box prompting demonstrating high accuracy on novel medical image segmentation tasks. We find that by using SAM, expert users can achieve fast semi-automatic segmentation of most relevant structures, making it highly valuable for interactive medical segmentation pipelines.

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
