# OpenReview forum: "SAM.MD: Zero-shot medical image segmentation capabilities of the Segment Anything Model"
_MIDL.io/2023/Short_Paper_Track — MIDL 2023 Short paper track Poster_

### Official Review · Reviewer_MP61 · 2023-04-10
**This is timely application for SAM model**

**Rating:** 9
**Confidence:** 5

**Review:**

This is a very interesting MIDL short paper. It shows that SAM generalizes well to CT data, making it a potential catalyst for the advancement of semi-automatic segmentation tools for clinicians. However, the current performance is still lower than SOTA, which makes it an interesting discussion for MIDL conference.

---

### Official Review · Reviewer_TdiV · 2023-04-10
**The paper evaluates the Segment Anything Model (SAM) for medical image segmentation using point or bounding box prompts, and shows that SAM can generalize well to CT data and perform competitively compared to a trained 2D and 3D nnU-Net baseline.**

**Rating:** 6
**Confidence:** 3

**Review:**

The paper evaluates the Segment Anything Model (SAM) for medical image segmentation using point or bounding box prompts and shows that SAM can generalize well to CT data and perform competitively compared to a trained 2D and 3D nnU-Net baseline. SAM has the potential to be a highly potent starting point for further adaptations to the intricacies of the medical domain and can be useful for interactive segmentation pipelines, enabling rapid semi-automatic segmentation with only a few clicks or bounding box prompts by an expert. However, SAM may experience decreased accuracy in areas near class boundaries, and as such areas can be manually segmented, the use of SAM might still greatly improve the speed of a clinical pipeline while maintaining a good level of accuracy.

Pros:
The paper presents an initial evaluation of the Segment Anything Model (SAM) for medical image segmentation using few clicks and bounding box prompting, showing high accuracy on novel medical image segmentation tasks.
SAM demonstrates potential as a zero-shot technique for medical image segmentation, enabling rapid semi-automatic segmentation of structures of interest with only a few prompts by an expert.
SAM can generalize well to CT data, making it a potential catalyst for the advancement of semi-automatic segmentation tools for clinicians.
SAM allows a variety of prompts, including text, points, and boxes, enabling zero-shot semantic segmentation.
SAM's box prompting, even with moderate jitter, is highly competitive against the baselines compared to point prompts.
SAM is seen to segment novel target structures (organs) to highly competitive accuracies compared to the baselines.
SAM is beneficial for interactive segmentation pipelines, improving the speed of a clinical pipeline while maintaining a good level of accuracy.

Cons:
The technical development is limited because it uses existing tools.
SAM does not reach state-of-the-art segmentation performance in the authors' investigations.
SAM may experience decreased accuracy in areas near class boundaries.